# Latent Action Learning Requires Supervision in the Presence of Distractors

**Alexander Nikulin** [1 2]  **Ilya Zisman** [1 3 4]  **Denis Tarasov** [1]  **Nikita Lyubaykin** [1 5]  **Andrei Polubarov** [1 3 4]
**Igor Kiselev** [6]  **Vladislav Kurenkov** [1 5]

## Abstract

Recently, latent action learning, pioneered by Latent Action Policies (LAPO), have shown remarkable pre-training efficiency on observation-only data, offering potential for leveraging vast amounts of video available on the web for embodied AI. However, prior work has focused on distractor-free data, where changes between observations are primarily explained by ground-truth actions. Unfortunately, real-world videos contain action-correlated distractors that may hinder latent action learning. Using Distracting Control Suite (DCS) we empirically investigate the effect of distractors on latent action learning and demonstrate that LAPO struggle in such scenario. We propose LAOM, a simple LAPO modification that improves the quality of latent actions by **8x**, as measured by linear probing. Importantly, we show that providing supervision with ground-truth actions, as few as 2.5% of the full dataset, during latent action learning improves downstream performance by **4.2x** on average. Our findings suggest that integrating supervision during Latent Action Models (LAM) training is critical in the presence of distractors, challenging the conventional pipeline of first learning LAM and only then decoding from latent to ground-truth actions.

## 1. Introduction

Recently, a new wave of approaches based on latent action learning has emerged (Edwards et al., 2019), demonstrating superior pre-training efficiency on datasets without action labels in large-scale robotics (Ye et al., 2024; Chen et al., 2024b;a; Cui et al., 2024; Bruce et al., 2024) and reinforcement learning (Schmidt & Jiang, 2023). Latent Action Mod-

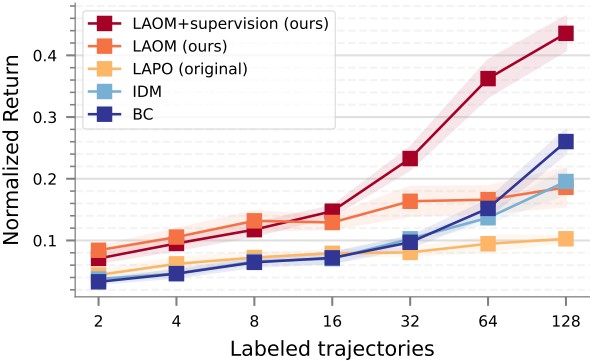

*Figure 1.* We show that in the presence of distractors, LAPO struggles to learn latent actions useful for pre-training and that simple BC or IDM are more effective. We propose LAOM, a simple modification that doubles the performance but still underperforms. Thus, we propose to reuse available ground-truth action labels to supervise latent action learning, which significantly improves the performance, achieving normalized score of 0.44. It recovers almost half the performance of BC with access to the full action-labeled dataset, while having access to only 2.5%. Results are averaged over four environments from Distracting Control Suite, three random seeds each. We provide per-environment plots on Figure 8. See Section 3 for the evaluation protocol, Sections 4 and 5 for method details.

els (LAM) infer latent actions between successive observations, effectively compressing observed changes. Under certain conditions, latent actions can even rediscover the ground truth action space (Schmidt & Jiang, 2023; Bruce et al., 2024). After training, LAM can be utilized for imitation learning on latent actions to obtain useful behavioral priors. For example, LAPA (Ye et al., 2024) showed that latent action learning can be used to pre-train large model on only human manipulation videos, and despite the huge cross-embodiment gap, still outperform OpenVLA (Kim et al., 2024b), which was pre-trained on expert in-domain data with available action labels.

Despite the initial success and the promise of unlocking *vast amounts of video available on the web* (Schmidt & Jiang, 2023; Ye et al., 2024), there is a critical shortcoming of previous work – it uses distractor-free data, where

[1]AIRI [2]MIPT [3]Skoltech [4]Research Center for Trusted Artificial Intelligence, ISP RAS [5]Innopolis University [6]Accenture. Correspondence to: Alexander Nikulin <nikulin@airi.net>. Work done by dunnolab.ai.

*Proceedings of the 42nd International Conference on Machine Learning*, Vancouver, Canada. PMLR 267, 2025. Copyright 2025 by the author(s).

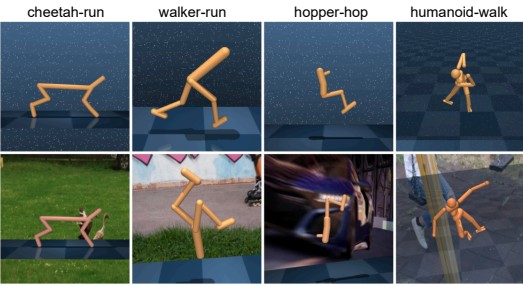

*Figure 2.* Visualization of the environments from the Distracting Control Suite (DCS) used in our work. Top row: without any distractors, identical to the original DeepMind Control Suite. Bottom row: with distractors, which consists of dynamic background videos, agent color change and camera shaking. See Section 3 for additional details.

all changes between observations are mainly and most efficiently explained by ground truth actions only, such as robot manipulation on a static background (Khazatsky et al., 2024). Unfortunately, this is not true for real-world web-scale data, as it contains a lot of action-correlated noise (Misra et al., 2024), e.g. people moving in the background. Such noise may better explain video dynamics and thus lead to latent actions unrelated to real actions. The phenomenon of overfitting to task-irrelevant information is not new and has been studied in model-based (Wang et al., 2024) and representation learning (Lamb et al., 2022; Zhang et al., 2020; Zhou et al., 2023). However, the effect of distractors on latent action learning, which we aim to address in this work, has not been similarly investigated.

In this work we empirically investigate the effect of action-correlated distractors on latent action learning using Distracting Control Suite (Stone et al., 2021). We demonstrate that naive latent action learning based on quantization and reconstruction objectives, such as LAPO (Schmidt & Jiang, 2023), struggle in the precense of distractors (see Section 4). We propose LAOM, a simple LAPO modification that improve the quality of latent actions by **8x**, as measured by linear probing, and double the downstream performance (see Figure 8). However, even after this, the resulting performance is only slightly better than simple Behavioral Cloning on available ground-truth actions. Thus, as our core contribution, we show that providing supervision with a small number, as little as 2.5% of the complete dataset, of action labels during LAOM training improves the downstream performance by **4.3x** on average (see Section 5), outperforming all baselines (see Figure 8). Our findings suggest that the pipeline used in most current work (Ye et al., 2024; Cui et al., 2024; Chen et al., 2024b) to first learn LAM and only then decode to ground-truth actions is suboptimal when distractors are present, as with supervision better result can be achieved using the same budget of actions labels. In addi-

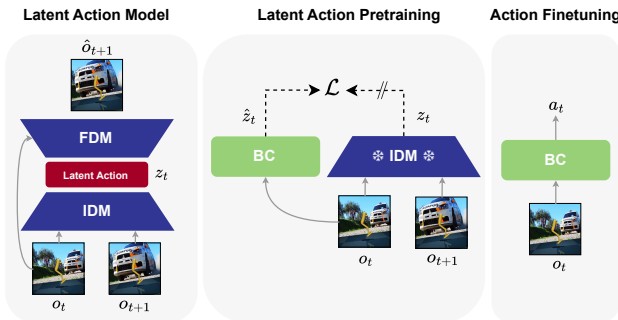

*Figure 3.* Overview of the latent action learning pipeline. In the first stage, the Latent Action Model (LAM) is pre-trained to infer latent actions between consecutive observations. In the second stage, the LAM is used to relabel the entire dataset with latent actions, which are then used for behavioral cloning. Finally, a decoder is trained to map from latent to true actions using a small number of labelled trajectories. In our work, we do not modify this pipeline in any way; we only examine the LAM architecture itself (see Figure 4).

tion, we show that latent action learning with supervision generalizes better in contrast to approaches based on inverse dynamics models (Baker et al., 2022; Zhang et al., 2022a; Zheng et al., 2023) but does not learn control-endogenous minimal state (Lamb et al., 2022).

## 2. Preliminaries

**Learning from observations.** Most methods in reinforcement learning require access to the dataset $\mathcal{D} := \{\tau_n\}_{n=1}^{N}$ of $N$ trajectories, where each $\tau_n := \{(o_i^n, a_i^n, r_i^n)\}_{i=1}^{\tau}$ contains observations, actions and rewards. Similarly, imitation learning requires access to trajectories $\tau_n := \{(o_i^n, a_i^n)\}_{i=1}^{\tau}$ that contain actions. Unfortunately, most expert demonstrations in the real world, such as YouTube videos of some human activity (Aytar et al., 2018; Baker et al., 2022; Zhang et al., 2022a; Ghosh et al., 2023), do not include rewards or action labels. Thus, researchers are actively exploring how to most effectively use the data $\tau_n := \{(o_i^n)\}_{i=1}^{\tau}$ without action labels to accelerate the learning of embodied agents at scale (Torabi et al., 2019). Still, we can often assume that a very small number of action labels are available. For example, previous work has explored ratios of up to 10% (Zheng et al., 2023), whereas in our work we allow a maximum of $\sim 2.5\%$ of labeled transitions.

**Latent action learning.** Latent action learning approaches (Edwards et al., 2019; Schmidt & Jiang, 2023; Chen et al., 2024b; Cui et al., 2024; Ye et al., 2024) aim to infer latent actions $z_t$ such that they are maximally informative about each observed transition $(o_t, o_{t+1})$ while being minimal. After the latent action model (LAM) is pre-trained, we can train policies to imitate latent actions on full data to obtain

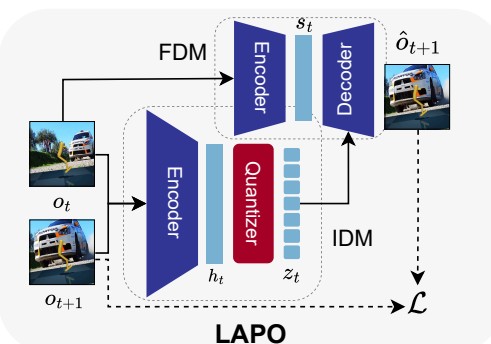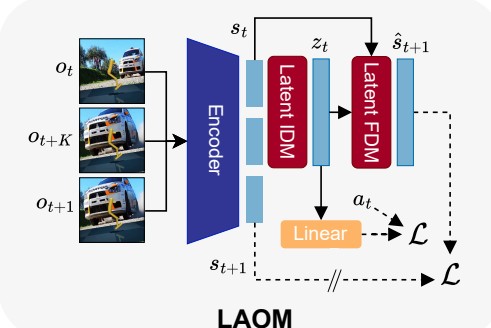

Figure 4. Simplified architecture visualization of LAPO, and LAOM - our proposed modification. LAPO consists of IDM and FDM, both with separate encoders, uses latent action quantization and predict next observation in image space via the decoder in FDM. LAOM incorporates multi-step IDM, removes quantization and does not reconstruct images, relying on latent temporal consistency loss. Images are encoded by shared encoder, while IDM and FDM operate in compact latent space. When small number of ground-truth action labels is available, we use them for supervision, linearly predicting from latent actions. For detailed description see Section 4.

useful behavioral priors. Finally, small decoder heads can be learned from latent to real actions of domain of interest.

We base our work on LAPO (Schmidt & Jiang, 2023), which is used in recent work (Chen et al., 2024b; Cui et al., 2024; Ye et al., 2024). LAPO uses two models in combination to infer latent actions. First is inverse dynamics model (IDM), which is given two consecutive observations predicts latent action $z_t \sim p_{\text{IDM}}(\cdot | o_t, o_{t+1})$. Second is forward dynamics model (FDM), which observes current observation and latent action, and predicts the next observation $\hat{o}_{t+1} \sim p_{\text{FDM}}(\cdot | o_t, z_t)$. Both models are trained jointly to minimize the next observation prediction loss $\|\hat{o}_{t+1} - o_{t+1}\|^2$. We illustrate the model architecture in Figure 4.

Given the information bottleneck on latent actions, e.g. quantization via the VQ-VAE (Van Den Oord et al., 2017), IDM cannot simply copy the next observation into the FDM as is, so it will be forced to compress and encode the difference between observations to be most predictive of the next observation. Without the distractors, through simplicity bias (Shah et al., 2020), the latent actions will recover the ground truth actions as they are most predictive of the dynamics. However, due to the presence of distractors, it may be not true for real-world data. In this work, we empirically examine how well current LAMs can recover true actions in such circumstances.

**Control-endogenous minimal state.** Lamb et al. (2022) defines *control-endogenous minimal state* as a representation that contains all the information necessary to control the agent, while completely discarding all irrelevant information. Lamb et al. (2022); Levine et al. (2024), theoretically and practically show that to learn such minimal state multi-step IDM should be used, i.e. IDM that predicts action

$a_t$ from states $s_t$ and $s_{t+k}$, where $k \in \{1, 2, 3, \ldots, K\}$. However, as showed by Misra et al. (2024), in the presence of *exogenous noise*, i.e. non-iid noise that is temporally action-correlated, the sample complexity of learning control-endogenous minimal state from video data can be exponentially worse than from action-labeled data. They hypothesized that this is true for latent action learning as well but did not provide any analysis regarding the quality of latent actions, which we tried to empirically address in this work.

## 3. Experimental Setup

**Environments and datasets.** To decouple the effects of latent action quality and exploration on performance, we work in an offline setting. For our purposes, it is *essential that the Behavior Cloning (BC) agent should recover most of the expert performance when trained on the full dataset with ground-truth actions revealed*, otherwise it would be difficult to understand the effect of latent action quality on pre-training.

As currently existing benchmarks with distractors (Stone et al., 2021; Ortiz et al., 2024) are not yet solved, we collect new datasets with custom difficulty, based on Distracting Control Suite (DCS) (Stone et al., 2021). DCS uses dynamic background videos, camera shaking and agent color change as distractors (see Figure 2 for visualization). The complexity is determined by the number of videos as well as the scale for the magnitude of the camera and the color change. We empirically found that using 60 videos and a scale of 0.1 is the hardest setting when BC can still recover expert performance. We collect datasets with five thousand trajectories for four tasks: cheetah-run, walker-run, hopper-hop and humanoid-walk, listed in the order of increasing

difficulty. See Appendix C for additional details.

**Evaluation.** To access the quality of the latent actions, we use two methods. First, we follow the approach of Zhang et al. (2022b) and use linear probing (Alain, 2016), which is a common technique used to evaluate the quality of learned representations by training a simple linear classifier or regressor on top the representations. Since we include ground truth actions in our datasets for debugging purposes, we train linear probes to predict them from latent actions simultaneously with the main method, e.g. LAPO (Schmidt & Jiang, 2023). We do not pass the gradient through the latent actions, so this does not affect the training. Second, following the most commonly used three-stage pipeline (Schmidt & Jiang, 2023; Chen et al., 2024b; Ye et al., 2024), we first pre-train LAM, then train BC model to predict latent actions on the full dataset, and finally, we reveal a small number of labeled trajectories to train a small two-layer MLP decoder from latent to real actions (see Figure 3). Using this decoder, we then evaluate the resulting agent in the environment for 25 episodes. To access scaling properties with different budgets of real actions, similar to Schmidt & Jiang (2023), we repeat this process for a variable number of labeled trajectories, from 2 to 128. All experiments are averaged over three random seeds.

**Baselines.** We use BC on true actions as our main baseline, since the main goal of latent action learning is to pre-train useful behavioral policies (Edwards et al., 2019; Schmidt & Jiang, 2023), which can be achieved by recovering true actions as accurately as possible. We use it in two ways. First, we try to get the best performance for each full dataset with true actions to use the final return for normalization. With such normalization, we can quantify how much performance we have recovered compared to if we had access to a fully action-labeled dataset. Second, we train BC from scratch on the same number of labels available to LAM, to evaluate the benefit of pre-training on large unlabeled data. Our last baseline is IDM, as it remains one of the most successful and simplest approaches to learn from action-free data at scale (Baker et al., 2022; Zhang et al., 2022a; Zheng et al., 2023). For additional details, see Appendix E.

We do not consider other possible types of unsupervised pre-training, as it was already extensively explored by other researchers (Tomar et al., 2021; Zhang et al., 2022b; Kim et al., 2024a), even with distractors (Misra et al., 2024; Ortiz et al., 2024). Our aim is not to compare latent action learning with existing approaches, but to investigate whether it works at all in the presence of action-correlated distractors.

**On hyperparameters tuning.** We tune the hyperparameters based on online performance for BC, on MSE to real actions on the full dataset for IDM, and on final linear probe MSE to real actions for latent action learning. In more practical tasks, we usually do not have this luxury, but since we are

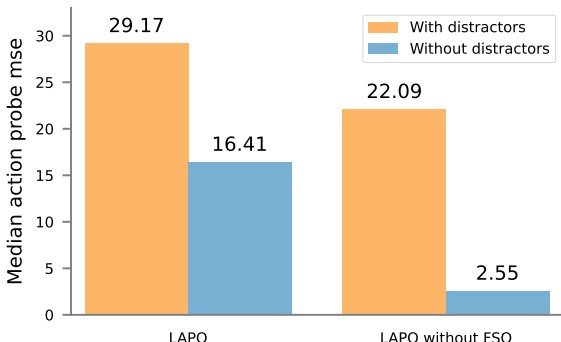

*Figure 5.* Quality of latent actions learned by LAPO. We show that quantization of latent actions significantly reduces the quality of actions, even on data without distractors, where LAPO should work without problems. Removing the quantization recovers the latent action quality, but additional modifications are needed to improve LAPO performance with distractors. Results are averaged across all four environments, each with three random seeds.

interested in estimating the upper bound performance of each method in a controlled setting, we believe that it is appropriate. For exact hyperparameters see Appendix F.

## 4. Latent Action Learning Struggle in the Presence of Distractors

To access the effect of distractors on latent action learning we start by carefully reproducing and adapting LAPO (Schmidt & Jiang, 2023) for our domain. We use similar architecture (see Figure 4) with ResNet (He et al., 2016) as observation encoders, borrowed from the open-source official LAPO implementation. Similar to Schmidt & Jiang (2023) we resize observations to 64 height and width, stacking 3 consecutive frames.

**Quantization hinders latent action learning.** To validate our implementation, we first measured performance on distractor-free datasets, which should not cause any difficulty. Contrary to previous research (Schmidt & Jiang, 2023; Chen et al., 2024b; Ye et al., 2024; Bruce et al., 2024), we found that commonly used latent action quantization during training significantly hindered the resulting latent action quality. We initially hypothesized that the problem might be with the VQ-VAE used for quantizing. In conversation with Schmidt & Jiang (2023) we confirmed that VQ-VAE is indeed susceptible to codebook collapse and requires extensive tuning. We tried the more modern FSQ (Mentzer et al., 2023), which has already been used successfully in RL (Scannell et al., 2024) and does not suffer from codebook collapse. Unfortunately, even after tuning, we were unable to improve the results significantly, so we simply removed it. To our surprise, this resulted in a large positive improvement (see Figure 5), but only for datasets

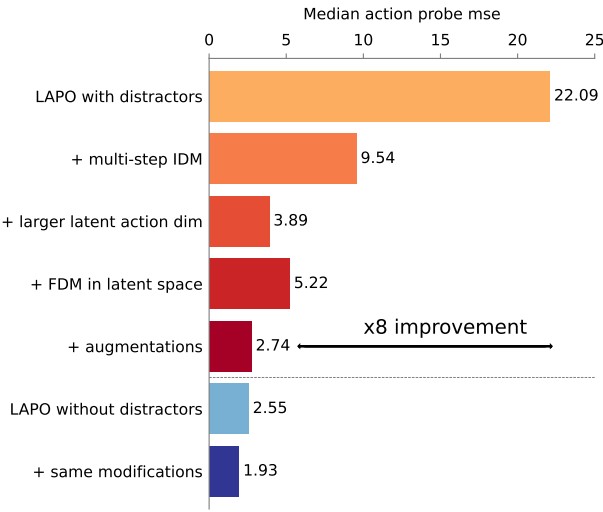

Median action probe mse

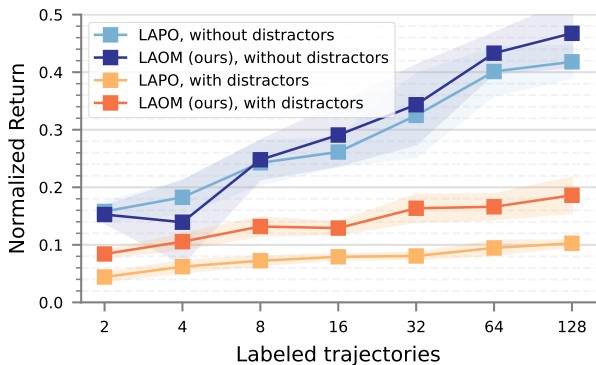

*Figure 6.* The individual effect of each proposed change in LAOM, our modification of LAPO, which overall improves latent action quality in the presence of distractors by a factor of 8. We describe the proposed changes in detail in Section 4 and visualize the final architecture in Figure 4. Results are averaged across all four environments, each with three random seeds.

without distractors, while with distractors the action quality remained at almost the same level.

One explanation for the result on Figure 5 may be that we are working with continuous actions, unlike the Schmidt & Jiang (2023) which used discrete actions. However, we believe that there are more general reasons. The main motivation for quantizing latent actions was to prevent shortcut learning, i.e. IDM copying $o_{t+1}$ to FDM as is, and to incentivize IDM to learn simpler latents that capture only action-related changes. We observed no evidence for shortcut learning, suggesting that it is unlikely to occur with high-dimensional observations, similar to the unlikelihood of collapse in Siamese networks (Chen & He, 2021). More importantly, in the presence of action-correlated distractors, *the information bottleneck may have the opposite effect, incentivising the IDM to encode noise into latent actions*. This noise can explain the dynamics more easily, so without guidance, the IDM has no way of distinguishing it from real actions. Therefore, we advise against the use of quantization for LAM training on real-world data.

**Latent action quality can be significantly improved.** As Figure 5 shows, naive LAPO may not be able to learn good latent actions in the presence of distractors and further improvements are needed. Thus, we propose simple modifications to the LAPO architecture, which in combination improve latent action quality by **8x**, almost closing the gap with distractor-free setting (see Figure 6). Interestingly, on distractor-free data improvements are marginal, further

*Figure 7.* Performance evaluation of the LAPO and the proposed LAOM with and without distractors. As can be seen, large gap in performance remains in the presence of distractors. Results are averaged across all four environments, each with three random seeds.

demonstrating the importance of the proposed changes to specifically help latent action learning in the presence of distractors. We visualize the resulting architecture, which we called Latent Action Observation Model (**LAOM**) in Figure 4 and describe changes in detail next:

*Multi-step IDM.* Inspired by research on control-endogenous minimal state discovery (Lamb et al., 2022; Levine et al., 2024) via multi-step IDM, we slightly modify our IDM objective to estimate latent action $z_t$ from $o_t$ and $o_{t+k}$, where $k \in \{1, 2, 3, \ldots, K\}$, instead of just consecutive observations. During training, we sampled $k$ uniformly for each sample and found that $K := 10$ worked best. Multi-step IDM helps to learn representation which encodes control-endogenous information with respect to current latent actions, which in turn helps learn better latent actions. This simple change alone doubled the latent action quality.

*Increasing latent actions capacity.* So far we have used latent actions with 128 dimensions, as in the original LAPO. However, for reasons similar to quantization removal, we significantly increased it to 8192, as it allows better next-observation prediction. Since IDM cannot distinguish control-related features from noise, the best we can hope for in general is to learn the full dynamics of the environment as accurately as possible. In such a case, latent actions will by definition contain true actions and we will be able to extract them via the probe. This change gives an additional 2.5x improvement.

*Removing observation reconstruction.* The need to fully reconstruct the next observation forces latent actions to encapsulate changes in each pixel, which is not always related to true actions, e.g. video in the background. Thus, we use the latent temporal consistency loss (Schwarzer et al., 2020; Hansen et al., 2022; Zhao et al., 2023) to predict next

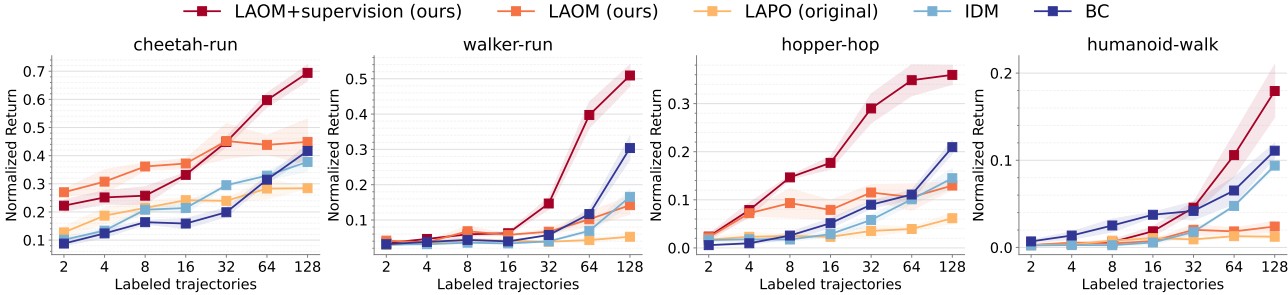

*Figure 8.* Performance evaluation of latent action learning approaches and baselines across different budgets of ground-truth action labels. As can be seen, LAPO struggles in the presence of distractors, being outperformed by simpler baselines. LAOM, our modification of LAPO, performs better, but not significantly. However, when we reuse the same labels used for decoding from latent to true actions to provide supervision during LAOM training (see Section 5), we significantly improve downstream performance, outperforming baselines in all environments. Importantly, all methods were pre-trained on the same unlabeled datasets and had access to exactly the same action labels, differing only in their use. Results are averaged over three random seeds. For a detailed description of the evaluation pipeline, see Section 3.

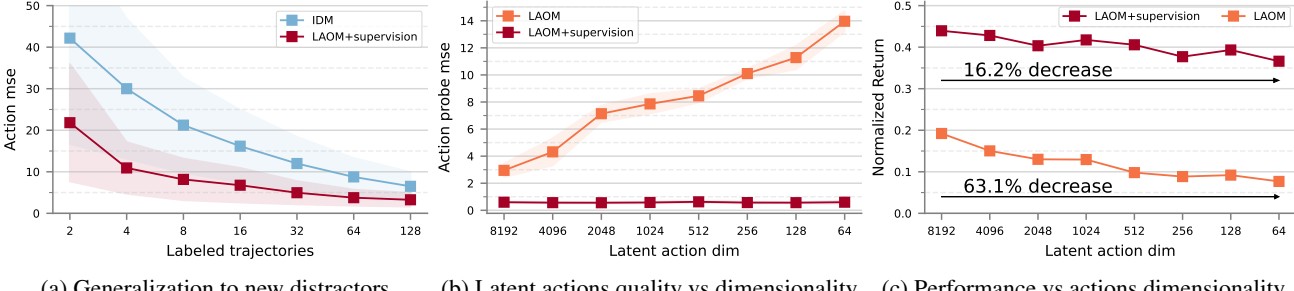

| (a) Generalization to new distractors | (b) Latent actions quality vs dimensionality | (c) Performance vs actions dimensionality |

*Figure 9.* (a) We show that latent action learning with supervision generalizes better than IDM to novel distractors for all considered budgets of ground-truth action labels available for pre-training. (b)-(c) Supervision with a small number of ground-truth actions during latent action learning allows for smaller action dimensionality without major performance degradation. Without supervision, the quality of latent actions, as well as performance, quickly degrades.

observation in compact latent space without reconstruction. IDM and FDM now operate on latent representation and consist of MLPs instead of ResNets (see Figure 4). This brings additional benefits, as with such architecture we can get rid of expensive decoder, reducing model size and increasing training speed. For target next observation we use simple stop-grad as in Chen & He (2021) or EMA encoder (Schwarzer et al., 2020). These change alone slightly increases probe MSE due to the instabilities. We fix them with the next change.

*Adding augmentations.* Augmentations are commonly used in conjunction with self-supervised objectives to stabilize training and avoid collapse (Schwarzer et al., 2020; Hansen et al., 2022; Zhao et al., 2023). Similarly, we found that augmentations help with stability and improve performance to even smaller probe MSE. We use the subset of augmentations from Almuzairee et al. (2024), which consists of random shifts, rotations and changes of perspective. We apply then only during latent actions training and do not use

in later stages.

**The large gap in downstream performance remains.** As Figure 7 shows, our improvements partially transfer to downstream performance, as LAOM outperforms vanilla LAPO on all label budgets, improving performance by up to 2x. LAOM also outperforms LAPO on data without distractors, but not significantly. However, there remains a large gap in final performance with and without distractors. We should emphasize that this gap is not due to the fact that setting with distractors is more difficult for BC, for example. We normalize performance by the return achieved by BC trained on each full dataset with ground-truth actions. Thus, the difference in performance is relative to BC and is explained by a difference in the quality of the latent actions.

Unfortunately, linear probing has a major limitation - it can only tell us whether real actions are contained in latent actions or not. For example, by increasing the dimensionality of latent actions in LAOM, we have improved the quality

according to the probe, but sacrificed their minimality, i.e. they additionally describe full dynamics, that is mostly unrelated to real actions. This can be detrimental as, during the BC stage, not only do we waste capacity predicting actions with higher dimensionality, but we also risk learning spurious correlations. This is probably the main reason for the poor performance, but it is the best we can do, otherwise latent actions will not contain true actions at all.

# 5. Latent Action Learning Requires Supervision

In previous sections, we proposed LAOM, an improved version of LAPO which almost doubled the downstream performance in the presence of distractors for all budgets of true action labels considered. However, overall performance remained quite low. Similar to unlikelihood of recovering the control-endogenous minimal state in the presence of distractors (Misra et al., 2024), our results suggest that without any supervision latent action learning may not be able to learn actions useful for efficient pre-training. What if we can provide supervision? Even the smallest number of true actions may ground latent action learning to focus on control-related features. We explore this in the following experiments.

**Supervision significantly increases downstream performance.** Despite the fact that existing approaches (Schmidt & Jiang, 2023; Ye et al., 2024; Chen et al., 2024b) pre-train LAM without true actions, in practice we still need to have some number of labels to learn the action decoder as last stage. We reuse these labels to provide supervision by linearly predicting them from latent actions during LAOM training (see Figure 4 for the final architecture). We plot the resulting downstream performance for each environment in Figure 8 and summarize in Figure 1. As can be seen, LAOM+supervision outperforms all baselines and scales better with a larger budget of real actions. It achieves an average normalized score of 0.44, i.e. it recovers almost half the performance of BC with access to the full dataset of true actions, while using only 2.5% of the action labels. Importantly, all methods have access to exactly the same number of action labels, differing only in how they use them. We provide results for distractor-free data in Appendix A.

**Latent action learning with supervision generalizes better that IDM.** Learning to predict true actions with IDM with a small number of labels and then relabeling larger datasets has recently been a quite successful approach (Baker et al., 2022; Zheng et al., 2023). Unfortunately, IDM is greatly limited in its generalization capabilites as dataset with labels may not contain some distractors or cover all actions. LAOM+supervision on other hand pre-trains on full combined dataset and can adapt better to larger variety of distractors and actions. We confirm this intuition in Fig-

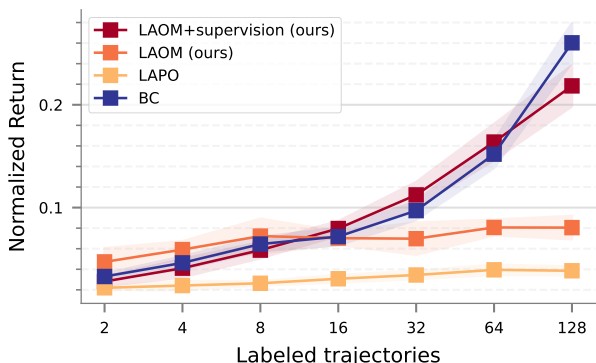

*Figure 10.* Evaluation of latent action learning approaches in cross-embodied pre-training in the presence of distractors, e.g. pre-training LAM on datasets from three environments and fune-tuning on action labeled data from the remaining one. Supervision during latent action pre-training improves downstream performance. However, overall performance is comparable to that of a simple BC trained from scratch on available action labels. Results are averaged across all four environments, each with three random seeds.

ure 9a measuring action prediction accuracy on evaluation dataset with never seen distractor background videos. IDM indeed generalizes worse than LAOM+supervision.

**Supervision enables compact latent actions without large performance degradation.** As we mentioned earlier very high dimensional latent actions are not optimal, as they may not be minimal, i.e. contain control-unrelated information and require larger BC models to imitate accurately. Similarly, LAPA (Ye et al., 2024) also reported that more compact latent action space increases pre-training efficiency. Unfortunately, the effectiveness of LAPO and even LAOM degrades dramatically when the dimensionality of latent actions is reduced. In Figure 9b and Figure 9c we show that supervision can partially mitigate this effect. LAOM+supervision loses only 16% of performance when reducing latent actions dimensionality from 8192 to 64, compared to 63% loss for LAOM. We used 128 labeled trajectories for this experiment.

**Supervision improves cross-embodied pre-training.** So far we have used homogeneous datasets, which contain data from only one environment. However, in practice our hope is to pre-train LAM on large and diverse dataset from different embodiments, including humans (McCarthy et al., 2024; Ye et al., 2024). To access performance in such a scenario, we assemble cross-embodied datasets in a leave-one-out fashion, e.g. for the cheetah-run, we sample 1666 trajectories (to get $\sim$ 5k) from other environments and combine them into a single dataset. We pre-train LAM and BC on them as usual and use the labeled data from the excluded environment for action decoding or supervision during LAOM

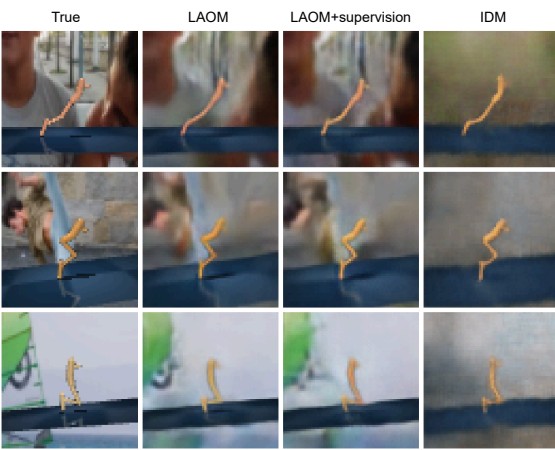

True    LAOM    LAOM+supervision    IDM

*Figure 11.* In contrast to IDM, latent action learning encode a lot of control-unrelated information, such as background videos, into the observation representations. This finding suggest that using latent action learning exclusively as a way to pre-train visual representations is not viable in the presence of distractors. We visualize the representations by training a separate decoder to reconstruct original observations.

training. As Figure 10 shows supervision during LAM pre-training yields a large performance improvement. However, the final performance is no better than training BC only on the provided labels from scratch. This is slightly concerning and further emphasizes the limitations of LAM methods in the presence of distractors.

**In contrast to IDM, latent action learning does not learn minimal state.** DynaMo (Cui et al., 2024) used latent action learning only as an objective to pre-train visual representations, not to obtain useful latent actions. To access the viability of such approach, we additionally train decoders to reconstruct original observations from the representations learned by LAM and IDM. What information does LAM encodes into its representations? As Figure 11 shows decoders were able to reconstruct original observations quite well, indicating that both LAOM and LAOM+supervision encode a lot of control-unrelated information, including distractors. In contrast, multi-step IDM truly learns control-endogenous minimal state as predicted by Lamb et al. (2022); Islam et al. (2022); Levine et al. (2024), fully ignoring control-unrelated information, such as background videos or agent color. This result appears to provide compelling evidence that using LAM exclusively as a way to obtain visual representations is not a viable approach in the presence of distractors.

## 6. Related Work

**Action relabeling with inverse dynamics models.** Simplest approach to utilize unlabeled data it to pretrain IDM on small number of action labels to further re-label a much

large dataset (Torabi et al., 2018). Baker et al. (2022) showed that this approach can work on a scale, achieving great success in Minecraft (Kanervisto et al., 2022). Zhang et al. (2022a) used similar pipeline, unlocking hours of in-the-wild driving videos for pretraining. Schmeckpeper et al. (2020) used unlabeled human manipulation videos within online RL loop, which supplied labels to IDM for re-labeling. Zheng et al. (2023) conducted large scale analysis of IDM re-labeling in offline RL setup, showing that only 10% of suboptimal trajectories with labels is enough to match performance on fully labeled dataset.

In contrast to previous work (Schmeckpeper et al., 2020; Baker et al., 2022; Zheng et al., 2023), we show that while IDM is a strong baseline in setups without distractors (see Figure 15 in Appendix A), it generalizes poorly when distractors are present. Our results show that when a small number of action labels are available, it is much better to combine IDM and latent action learning to achieve much stronger performance and generalization (see Figure 8), suggesting that for web-scale data (Baker et al., 2022; Zhang et al., 2022a) our approach may be better than simple IDM re-labeling.

**Latent action learning.** To our knowledge, Edwards et al. (2019) was the first to propose the task of recovering latent actions and *imitating latent policies from observation*, with limited success on simple problems. However, the original objective had scalability issues (Struckmeier & Kyrki, 2023). LAPO (Schmidt & Jiang, 2023) greatly simplified the approach, removed scalability barriers, and for the first time achieved high success on the hard, procedurally generated ProcGen benchmark (Cobbe et al., 2020). Latent action learning was further scaled by Bruce et al. (2024); Cui et al. (2024); Ye et al. (2024); Chen et al. (2024b;a) to larger models, data, and harder, more diverse robotics domains.

In contrast to our work, all the mentioned approaches (Schmidt & Jiang, 2023; Ye et al., 2024; Cui et al., 2024; Chen et al., 2024b;a) use data without distractors, where all changes in dynamics are mainly explained by ground truth actions only. As we show in our work (see Section 4), naive latent action learning does not work in the presence of distractors. Although we propose improvements that double the performance, it is not enough (see Figure 7). Providing supervision with a small number of action labels during LAM training significantly improves performance (see Figure 1), suggesting that the pipeline used in most current work (Ye et al., 2024; Cui et al., 2024; Chen et al., 2024b;a) to first learn LAM and only then decode to ground-truth actions is suboptimal.

The most closely related to us is the work of Cui et al. (2024), which also removes latent action quantization, the reconstruction objective in favor of latent temporal con-

sistency (Schwarzer et al., 2020; Zhao et al., 2023), and provides ablation with ground-truth actions supervision during LAM training. However, they train LAM only as a way to pre-train visual representations and do not provide any analysis regarding the effect of their proposed changes on the quality of the resulting latent actions. This also explains why they report that supervision with true actions gives no improvement, while we show that it gives significant gains (see Figure 1). Moreover, visually reconstructing representations, we show that latent action learning methods do not produce control-endogenous state (see Figure 11), and thus are probably not suitable as a method of visual representation learning in the presence of distractors.

## 7. Limitations

There are several notable limitations to our work. First, although we used the Distracting Control Suite (Stone et al., 2021), which allows us to precisely control the difficulty of distractors in a convenient way and clearly access generalization to new distractors, the overall distribution and noise patterns may be quite different compared to real-world videos on the web. Thus, our conclusions may not be fully applicable, e.g. it is possible that supervision is not as important for relevant to embodied AI data, or vice versa, it may turn out to be much more necessary for good results than we have used. Nevertheless, we believe that the overall conclusion about the need for some form of supervision is quite general.

Second, the need for supervision for latent action learning is a serious limitation, as compared to our setup, which is more reminiscent of Minecraft (Kanervisto et al., 2022) or Nethack (Hambro et al., 2022), where both labeled and unlabeled data are available, we have no chance to get real labels for already existing videos on the web or to fully cover their diversity with hand-crafted labels. Therefore, further research is needed to find out whether pre-training LAM on web data combined with supervision on robot data will achieve a similar effect, although our preliminary experiment on cross-embodied pre-training is pessimistic. It is quite possible that supervision can come in other forms than ground-truth actions, as we simply need a way to ground latent actions on control-related features of the observations. For example, for egocentric videos (Grauman et al., 2022) we can use hand tracking as a proxy action to supervise latent action learning.

Finally, similar to offline RL (Levine et al., 2020), the problem of hyperparameter tuning remains, since without action labels there is currently no way to access the quality of latent actions.

## 8. Conclusion

In this work, we empirically investigated the effect of action-correlated distractors on latent action learning. We showed that LAPO struggles to learn latent actions useful for pre-training. Although we proposed LAOM, a simple modification of LAPO, which doubled performance, it did not fully close the gap with the distractor-free setting. Crucially, we found that even minimal supervision - reusing as little as 2.5% of the dataset's ground-truth action labels during latent action learning significantly improved downstream performance, challenging the conventional pipeline of first pre-training LAM and only then decoding from latent to real actions. Our findings suggest that integrating supervision is essential for robust latent action learning in real-world scenarios, paving the way for unlocking the vast amounts of video data available on the web for embodied AI. We discuss the limitations of our work in the Section 7.

## Acknowledgements

This work was supported by the Ministry of Economic Development of the RF (code 25-139-66879-1-0003).

## Impact Statement

This paper presents work whose goal is to advance the field of Machine Learning. There are many potential societal consequences of our work, none which we feel must be specifically highlighted here.

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

## A. Additional Figures

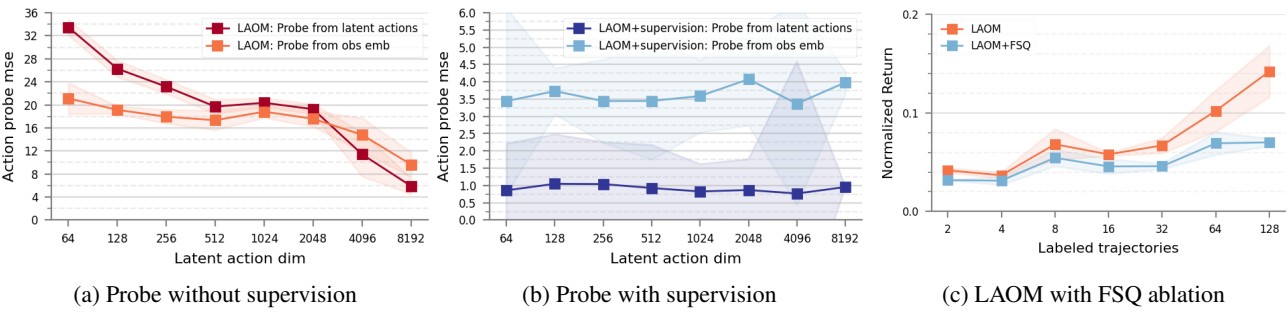

(a) Probe without supervision      (b) Probe with supervision      (c) LAOM with FSQ ablation

*Figure 12.* We provide additional ablations on the walker environment with three random seeds. (a)-(b) We took the observation embedding from the LAOM visual encoder and trained the linear probe to predict real actions, similar to probing from latent actions. (a) As can be seen, for LAOM the probe from observation embedding is better for smaller latent action dimensionality. This can be explained by the fact that the information bottleneck induces the IDM to mainly encode noise in latent actions, as it can better explain the dynamics (deterministic distractors in the background), while observation embedding mostly preserves the information. At higher latent action dimensions, they are expected to equalize, as latent actions without bottleneck can encode the full dynamics, including noise and real actions. This is exactly the effect we described in Section 4 which motivated us to add supervision. (b) However, we see a different effect with LAOM+supervision, where the probe from the embedding observation is generally worse than from the latent actions, as with supervision we can ground the latent actions to focus on features relevant for control even with small dimensions, filtering out the noise. (c) We re-evaluate the effect of the quantization during LAM training, given all other LAOM improvements from Section 4 and measuring actual performance in the environment instead of probing. As can be seen, quantization is indeed harmful for performance.

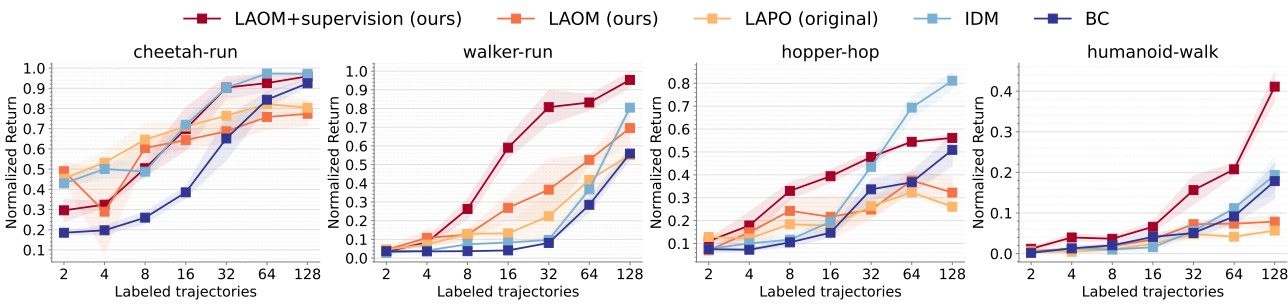

*Figure 13.* Main results without distractors, analogously to our main result in Figure 8. As can be seen, supervision help even without distractors, although all methods work good in this setting. Notably, IDM is a strong baseline.

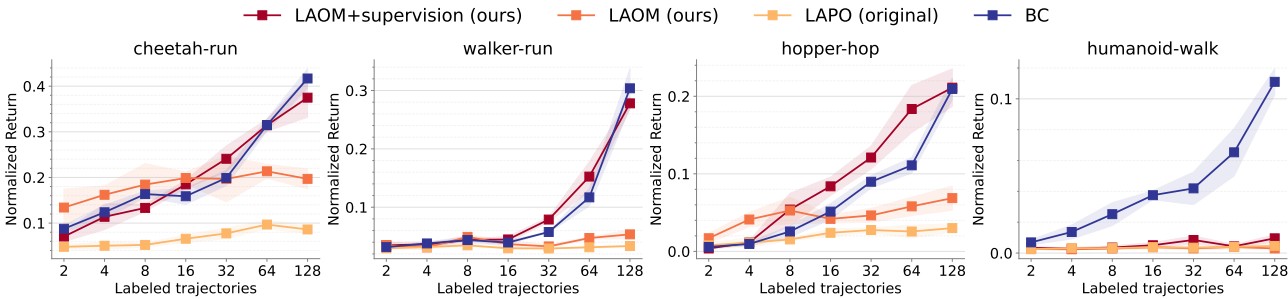

*Figure 14.* Mixed-embodied pre-training experiment results for each environment. For details see Figure 14.

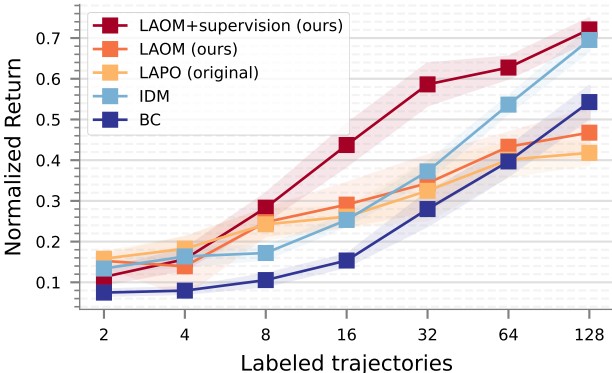

*Figure 15.* Figure summarizing the results from Figure 13, analogously to our main result in Figure 1. As can be seen, supervision help even without distractors, although all methods work good in this setting.

## B. Additional Related Work

**Learning with distractors.** Distractors in various forms are commonly used in many sub-fields of reinforcement learning, such as: visual model-based learning, model-free learning, and representation learning.

In model-based learning, researchers explore ways to efficiently train world models that do not waste their capacity to model task-irrelevant details, either via decomposing world models to predict relevant and irrelevant parts separately (Fu et al., 2021; Wang et al., 2022; Wan et al., 2023; Wang et al., 2024) or by avoiding reconstructing observations (Okada & Taniguchi, 2021; Deng et al., 2022; Zhu et al., 2023; Liu et al., 2023b; Burchi & Timofte, 2024). In our work, we have a similar need to not model action irrelevant details, as this will result in latent actions that describe changes in exogenous noise, not changes cased by ground truth actions. Thus, we use the commonly occurring latent temporal consistency loss (Schwarzer et al., 2020; Hansen et al., 2022; Zhao et al., 2023).

In model-free learning, researchers explore various techniques to improve generalization to new distractors and domain shifts (Hansen & Wang, 2021; Hansen et al., 2021; Bertoin et al., 2022; Huang et al., 2022b; Batra & Sukhatme, 2024; Almuzairee et al., 2024), which often revolves around the use of augmentations (Ma et al., 2022). In our work we also use augmentations, specifically a subset of ones proposed by Almuzairee et al. (2024), to stabilize LAM training with latent temporal consistency loss (Schwarzer et al., 2020).

In representation learning, researchers search for ways to obtain minimal representations that contain only task- (Yamada et al., 2022), reward- (Zhou et al., 2023) or control-related information (Zhang et al., 2020; Lamb et al., 2022; Liu et al., 2023a; Ni et al., 2024; Levine et al., 2024), as this can greatly increase sample efficiency and generalization (Kim et al., 2024a). In our work, inspired by Lamb et al. (2022), we incorporate the multi-step IDM into LAM and show that it can help learn better latent actions in the presence of exogenous noise. Moreover, when small number of ground truth actions is available for pre-training (see Figure 1), our model on them conceptually reduces to one proposed by Levine et al. (2024), for which it has been theoretically shown that it can recover control-endogenous minimal state. This may explain why incorporating labels during LAM pre-training, rather than during final fine-tuning, brings so much benefit, since discovering true actions is trivial given a minimal state. We however, found a contradicting evidence, as Figure 11 shows that our proposed methods do not learn minimal state in practice.

Overall, although we were inspired by existing approaches, they have not previously been used to improve latent action learning, especially in combination, which, as we show (see Figure 6) is essential for good performance in the presence of distractors.

## C. Data Collection

We used environments from the Distracting Control Suite (DCS), wrapped with Shimmy wrappers for compatibility with the Gymnasium API. For cheetah-run, walker-run and hopper-hop we used PPO (Schulman et al., 2017), adapted from

the CleanRL (Huang et al., 2022a) library. For humanoid-walk, we used SAC (Haarnoja et al., 2018) from the stable-baselines3 (Raffin et al., 2021) library, as PPO from CleanRL was not able to solve it at the expert level. We used default hyperparameters and trained on 1M transitions in each environment, except for humanoid-walk, where we trained on 100k transitions. Importantly, for speed, all experts were trained with proprioceptive states and no distractors, we later rendered proprioceptive states to 64px images with or without distractors during data collection. For each environment, we collected 5k trajectories, with an additional 50 trajectories for evaluation with novel distractor videos (from the evaluation set in the DCS). As each trajectory consists of 1000 steps, the datasets contain 5M transitions. We include ground truth actions and states for debugging purposes. The datasets will be released together with the main code repository.

*Table 1.* Datasets statistics.

| Dataset | Average Return | Size (GB) |
|---|---|---|
| cheetah-run | 837.70 | 57.7 |
| walker-run | 739.79 | 57.8 |
| hopper-hop | 306.63 | 57.6 |
| humanoid-walk | 617.22 | 58.9 |

## D. Implementation Details

All experiments were run on H100 GPUs, in single-gpu mode and PyTorch bf16 precision with AMP. For the visual encoder, we used ResNets from the open-source LAPO (Schmidt & Jiang, 2023) codebase, which also borrowed from baselines originally provided as part of the ProcGen 2020 competition. For the action decoder, we used a two-layer MLP with 256 hidden dimensions and ReLU activations.

In contrast to the commonly used cosine similarity, we used MSE for temporal consistency loss. We also found that projection heads degraded performance, so we did not use them. We use slightly non-standard MLP for latent IDM and FDM: we compose it from multiple MLP blocks inspired by Transformer architecture (Vaswani, 2017) and condition on latent action and observation representation on all layers instead of just the first. We have found that this greatly improves prediction, especially for latent actions. We also use ReLU6 activations instead of GELU, as it naturally bounds the activations, which helps with stability during training, similar to target networks in RL (Bhatt et al., 2019). Without supervision, we use the EMA target encoder. With supervision, we find that a simple stop-grad is sufficient to prevent any signs of collapse, a finding also reported by Schwarzer et al. (2020).

For all experiments we use the cosine learning late schedule with warmup. For hyperparameters see Appendix F. We open-source the code at `https://github.com/dunnolab/laom`.

*Table 2.* Methods training time summed from all stages (including online evaluation) for each method.

| Method | Training Time |
|---|---|
| LAPO | $\sim$ 7h 38m |
| LAOM | $\sim$ 6h 43m |
| LAOM+supervision | $\sim$ 7h 6m |
| BC | $\sim$ 1h 10m |
| IDM | $\sim$ 5h 30m |

## E. Evaluation Details

We outline the evaluation procedures used in our experiments for each method. First, we review the general setup. For each environment, we have a large dataset without action labels, with and without distractors. To decode the learned latent actions to ground truth for evaluation, we allow a small amount of action labeled data, in line with previous work (Schmidt & Jiang, 2023; Ye et al., 2024). We sample it once from the existing dataset, revealing true actions, to ensure that all methods are on equal conditions. We use identical backbones where possible, and try our best to make all methods equal in the number of

trainable weights. For hyperparameters, see Appendix F. We report the scores achieved by BC trained on datasets with all actions revealed in Table 4. We use these for normalization in all our experiments.

**BC.** We trained BC from scratch to predict ground-truth actions on available labels, i.e. on 2 or 128 trajectories.

**IDM.** We used two-staged pipeline. First, we trained IDM to predict actions on available labels, i.e. on 2 trajectories. Then, we trained BC on full unlabeled dataset, providing labels via pre-trained IDM. We report BC final return.

**LAPO and LAOM.** We used three-stage pipline. First, we pre-train latent actions on full unlabeled datasets. Then, we trained BC, providing latent action labels via pre-trained LAM. Finally, we trained action decoder on small amount of labels, while freezing the rest of the policy weights.

**LAOM+supervision.** Almost like LAOM, with the difference being that we exactly aligned stages in terms of action labels used. While in LAOM we can pre-train it once and then re-use for later stages regardless of the number of action labels, in LAOM+supervision we trained separate LAM for each budget of labels. Thus, for LAOM+supervision trained with supervision from 32 trajectories of labels, on final stage the decoder was trained only on the same 32 trajectories. We repeat this process for all cases, from 2 to 128 trajectories.

*Table 3.* Evaluation returns of BC trained on full datasets with ground-truth actions revealed. We use them for normalization.

| Dataset | With distractors | Without distractors |
|---|---|---|
| cheetah-run | 823 | 840 |
| walker-run | 749 | 735 |
| hopper-hop | 253 | 300 |
| humanoid-walk | 428 | 601 |

*Table 4.* Total parameters for each method according to the hyperparameters used in Appendix F.

| Dataset | Total Parameters |
|---|---|
| LAPO | 211847849 |
| LAOM | 192307136 |
| LAOM+supervision | 192479189 |
| BC (on all stages) | 107541504 |
| IDM | 192258965 |

# F. Hyperparameters

*Table 5.* LAPO hyperparameters. We use the same hyperparameters for all experiments and explicitly mention any exceptions. Names are exactly follow the configuration files used in code.

| Stage | Parameter | Value |
|---|---|---|
| Latent actions learning | grad_norm | None |
| | batch_size | 512 |
| | num_epochs | 10 |
| | frame_stack | 3 |
| | encoder_deep | False |
| | weight_decay | None |
| | encoder_scale | 6 |
| | learning_rate | 0.0001 |
| | warmup_epochs | 3 |
| | future_obs_offset | 10 |
| | latent_action_dim | 8192 |
| | encoder_num_res_blocks | 2 |
| Latent behavior cloning | dropout | 0.0 |
| | use_aug | False |
| | batch_size | 512 |
| | num_epochs | 10 |
| | frame_stack | 3 |
| | encoder_deep | False |
| | weight_decay | None |
| | encoder_scale | 32 |
| | learning_rate | 0.0001 |
| | warmup_epochs | 0 |
| | encoder_num_res_blocks | 2 |
| Latent actions decoding | use_aug | False |
| | batch_size | 512 |
| | hidden_dim | 256 |
| | weight_decay | None |
| | eval_episodes | 25 |
| | learning_rate | 0.0003 |
| | total_updates | 2500 |
| | warmup_epochs | 0.0 |

*Table 6.* LAOM hyperparameters. We use the same hyperparameters for all experiments and explicitly mention any exceptions. Names are exactly follow the configuration files used in code.

| Stage | Parameter | Value |
|---|---|---|
| Latent actions learning | use_aug | True |
| | grad_norm | None |
| | batch_size | 512 |
| | num_epochs | 10 |
| | target_tau | 0.001 |
| | frame_stack | 3 |
| | act_head_dim | 1024 |
| | encoder_deep | False |
| | obs_head_dim | 1024 |
| | weight_decay | None |
| | encoder_scale | 6 |
| | learning_rate | 0.0001 |
| | warmup_epochs | 3 |
| | encoder_dropout | 0.0 |
| | act_head_dropout | 0.0 |
| | encoder_norm_out | False |
| | obs_head_dropout | 0.0 |
| | future_obs_offset | 10 |
| | latent_action_dim | 8192 |
| | target_update_every | 1 |
| | encoder_num_res_blocks | 2 |
| Latent behavior cloning | dropout | 0.0 |
| | use_aug | False |
| | batch_size | 512 |
| | num_epochs | 10 |
| | frame_stack | 3 |
| | encoder_deep | False |
| | weight_decay | None |
| | encoder_scale | 32 |
| | learning_rate | 0.0001 |
| | warmup_epochs | 0.0 |
| | encoder_num_res_blocks | 2 |
| Latent actions decoding | use_aug | False |
| | batch_size | 512 |
| | hidden_dim | 256 |
| | weight_decay | None |
| | eval_episodes | 25 |
| | learning_rate | 0.0003 |
| | total_updates | 2500 |
| | warmup_epochs | 0 |

*Table 7.* LAOM+supervision hyperparameters. We use the same hyperparameters for all experiments and explicitly mention any exceptions. Names are exactly follow the configuration files used in code.

| Stage | Parameter | Value |
|---|---|---|
| Latent actions learning | use_aug | True |
| | grad_norm | None |
| | batch_size | 512 |
| | num_epochs | 10 |
| | target_tau | 0.001 |
| | frame_stack | 3 |
| | act_head_dim | 1024 |
| | encoder_deep | False |
| | obs_head_dim | 1024 |
| | weight_decay | 0.0 |
| | encoder_scale | 6 |
| | learning_rate | 0.0001 |
| | warmup_epochs | 3 |
| | encoder_dropout | 0.0 |
| | act_head_dropout | 0.0 |
| | encoder_norm_out | False |
| | obs_head_dropout | 0.0 |
| | future_obs_offset | 10 |
| | labeled_loss_coef | 0.01 (0.001, cheetah-run) |
| | latent_action_dim | 8192 |
| | labeled_batch_size | 128 |
| | target_update_every | 1 |
| | encoder_num_res_blocks | 2 |
| Latent behavior cloning | dropout | 0.0 |
| | use_aug | False |
| | batch_size | 512 |
| | num_epochs | 10 |
| | frame_stack | 3 |
| | encoder_deep | False |
| | weight_decay | None |
| | encoder_scale | 32 |
| | learning_rate | 0.0001 |
| | warmup_epochs | 0 |
| | encoder_num_res_blocks | 2 |
| Latent actions decoding | use_aug | False |
| | batch_size | 512 |
| | hidden_dim | 256 |
| | weight_decay | 0 |
| | eval_episodes | 25 |
| | learning_rate | 0.0003 |
| | total_updates | 2500 |
| | warmup_epochs | 0 |

*Table 8.* IDM hyperparameters. We use the same hyperparameters for all experiments and explicitly mention any exceptions. Names are exactly follow the configuration files used in code.

| Stage | Parameter | Value |
|---|---|---|
| IDM learning | use_aug | False |
| | grad_norm | None |
| | batch_size | 512 |
| | frame_stack | 3 |
| | act_head_dim | 1024 |
| | encoder_deep | False |
| | weight_decay | None |
| | encoder_scale | 12 |
| | learning_rate | 0.0001 |
| | total_updates | 10000 |
| | warmup_epochs | 3 |
| | encoder_dropout | 0.0 |
| | act_head_dropout | 0.0 |
| | future_obs_offset | 1 |
| | encoder_num_res_blocks | 2 |
| Behavior cloning on IDM actions | dropout | 0.0 |
| | use_aug | False |
| | batch_size | 512 |
| | num_epochs | 10 |
| | frame_stack | 3 |
| | encoder_deep | False |
| | weight_decay | None |
| | encoder_scale | 32 |
| | eval_episodes | 25 |
| | learning_rate | 0.0001 |
| | warmup_epochs | 0 |
| | encoder_num_res_blocks | 2 |

*Table 9.* BC as baseline hyperparameters. We use the same hyperparameters for all experiments and explicitly mention any exceptions. Names are exactly follow the configuration files used in code.

| Parameter | Value |
|---|---|
| dropout | 0.0 |
| use_aug | false |
| batch_size | 512 |
| frame_stack | 3 |
| encoder_deep | false |
| weight_decay | 0 |
| encoder_scale | 32 |
| eval_episodes | 25 |
| learning_rate | 0.0001 |
| total_updates | 10000 |
| warmup_epochs | 0 |
| cooldown_ratio | 0 |
| encoder_num_res_blocks | 2 |

*Table 10.* BC for normalization hyperparameters. We use the same hyperparameters for all experiments and explicitly mention any exceptions. Names are exactly follow the configuration files used in code.

| Parameter | Value |
|---|---|
| dropout | 0.0 |
| use_aug | false |
| batch_size | 512 |
| frame_stack | 3 |
| encoder_deep | false |
| weight_decay | 0 |
| encoder_scale | 32 |
| eval_episodes | 25 |
| learning_rate | 0.0001 |
| num_epochs | 10 |
| warmup_epochs | 0 |
| cooldown_ratio | 0 |
| encoder_num_res_blocks | 2 |

