# OpenReview forum: "Latent Action Learning Requires Supervision in the Presence of Distractors"
_ICML.cc/2025/Conference — ICML 2025 poster_

### Official Review · Reviewer_nu2R · 2025-03-05

**Overall Recommendation:** 3

**Summary:**

This paper presents an empirical study of latent action learning in the presence of distractors. They found that latent action learning struggles with distractors, and propose several changes in architecture to improve latent action learning. Notably, they found supervision with a small amount of action labels could significantly improve latent action learning with distractors.

**Claims And Evidence:**

I am curious about the claim on "Quantization hinders latent action learning". it seems that this is only verified by linear probing. However, as the author mentioned, linear probing has a major limitation - it can only tell us whether real actions are contained in latent actions or not. Removing quantization, in some sense, is similar to increasing the dimensionality. As a result, I believe it would be better if the author could study this claim with experiments on the BC stage.

**Essential References Not Discussed:**

The paper fails to cite the following work on latent action learning that also introduce some improvements, such as using random cropping on data—an approach similar to the "adding augmentation" method discussed in the submission.

"IGOR: Image-GOal Representations are the Atomic Control Units for Foundation Models in Embodied AI"

**Experimental Designs Or Analyses:**

yes. seems sound to me

**Methods And Evaluation Criteria:**

Yes. It follows the experiment setting proposed in LAPO.

**Other Comments Or Suggestions:**

No.

**Other Strengths And Weaknesses:**

Overall, I appreciate the paper as an empirical study on latent action learning with distractors. The problem setting is realistic, and the experimental work is solid.

One limitation is that the study focuses solely on LAPO without comparing it to other latent action learning methods such as Genie, LAPA, IGOR, and MotoGPT, which utilize varying neural network architectures for the latent action model. However, this appears to be an inherent challenge in latent action research—each work introduces similar yet distinct latent action models tailored for different applications. As a result, direct comparisons between existing approaches are limited, making it difficult to determine which neural network architecture represents the state-of-the-art.

**Questions For Authors:**

I'm curious about the results shown in Figure 10. How is the separate decoder trained? Is the distractor constant, or does it display temporal correlations within each trajectory? If that's the case, then even when using only true actions, one might expect that an FDM decoder could learn to predict the distractor. Additionally, I'm not surprised that the LAM model is able to predict distractors, since it is trained by reconstructing the next frame, which naturally extends to predicting distractors as well.

**Relation To Broader Scientific Literature:**

the claim on line 126~127 is not very accurate. The NN architecture in LAPO is not used in motoGPT, Dynamo, and LAPA. I believe they use different architectures.

**Theoretical Claims:**

no theoretical claim

---

> ### Author Rebuttal · Authors · 2025-04-01
>
> We thank the reviewer for their time and effort. We address the questions below.
>
> > I am curious about the claim on "Quantization hinders latent action learning". it seems that this is only verified by linear probing. However, as the author mentioned, linear probing has a major limitation - it can only tell us whether real actions are contained in latent actions or not. Removing quantization, in some sense, is similar to increasing the dimensionality. As a result, I believe it would be better if the author could study this claim with experiments on the BC stage.
> >
>
> You are absolutely right that removing quantization has a similar effect to increasing the dimensionality of latent actions, as both changes loosen the information bottleneck, which we believe can be harmful in the presence of distractors (for details see the last response to reviewer **sdKR**).
>
> However, we disagree with the conclusion about linear probes. As you rightly point out, probes can only tell us whether real actions are contained in latent actions or not. If probe loss is low, we cannot be sure that latent actions are minimal, only that they contain real actions. To test their true quality, we need to pre-train BC and fine-tune it in a real environment. Thus, low probe loss is a necessary (but not sufficient) condition for good latent actions. This means that if we get a high probe loss, it definitely means that the latent actions do not contain real actions and are therefore useless for subsequent fine-tuning. This is the reason why we did not consider FSQ further in the later stages, as its addition only worsens the probe loss and therefore definitely worsens the final result.
>
> However, to be certain, we performed an experiment with LAOM+FSQ. Due to time constraints, we only used walker environment and 3 random seeds (see [Figure](https://ibb.co/VY0djs7T)). As can be seen, FSQ does indeed worsen the downstream performance after fine-tuning. We will include this Figure (and for the remaining environments) in the Appendix.
>
> > the claim on line 126~127 is not very accurate. The NN architecture in LAPO is not used in motoGPT, Dynamo, and LAPA. I believe they use different architectures.
> >
>
> You are right, we did not express ourselves clearly. In our case, the details of the architecture itself are not very important (even LAOM itself does not work well without supervision). What is important is that mathematically all these architectures do exactly the same thing as LAPO, and so inherit the same limitations. We will correct this statement in the new version of the paper.
>
> > The paper fails to cite the … "IGOR: Image-GOal Representations …"
> >
>
> Thank you for the suggestion, this is indeed a highly relevant paper. We will include the citation.
>
> > I'm curious about the results shown in Figure 10. How is the separate decoder trained? Is the distractor constant, or does it display temporal correlations within each trajectory? If that's the case, then even when using only true actions, one might expect that an FDM decoder could learn to predict the distractor.
> >
>
> For the decoder, we used an observation embedding after the ResNet encoder for each method. We trained it to reconstruct the observation and did not pass the gradients through the embedding to avoid changing the main training loop. The distractors are dynamic and change as the episode unfolds (video plays in the background, agent colour changes, camera shakes).
>
> You are right that with real actions, FDM from the original LAPO will learn to predict distractors. However, this is a problem of prediction in the pixel space. A method similar to LAOM, which predicts the next observation in latent space, with ground truth actions will provably recover the control endogenous minimal state, filtering out the distractors (see Preliminaries and [Multistep Inverse Is Not All You Need](https://arxiv.org/abs/2403.11940) paper). Why doesn't this happen with LAOM+supervision? Probably because the number of ground truth actions is extremely small in our case.

---

### Official Review · Reviewer_o8og · 2025-03-12

**Overall Recommendation:** 1

**Summary:**

Latent actions prove to be useful for efficient policy pretraining from unlabeled videos. This paper aims to enhance the quality of latent actions by removing the original information bottleneck, leveraging multi-step future observations, and predicting future states in the latent space. The authors also suggest that adding a small amount of action supervision can significantly mitigate the effects of distractors. The proposed methods are validated through linear probing accuracy and normalized returns in downstream tasks.

## *Update after rebuttal*

I remain unpersuaded by the authors' response. Specifically, while I agree that the modifications bring positive gains in the scenarios created by the authors, I still have the following major concerns:

- Due to the prediction loss and the extremely large latent dimension, the use of action supervision does not effectively produce a latent action representation. As revealed by Figure 10 and the responses to reviewers HSFB and sdKR, the latent capacity is so redundant that it passes almost all observation information through the latent. As a result, the policy is more like a video prediction policy rather than a latent action policy. However, the proposed method is not compared or discussed with previous video prediction policies [1,2].
- The conclusions may not be applicable to real-world applications (the authors have also mentioned this in Appendix A). The distractors are created by copying and pasting the background directly, which are quite different from real-world patterns. Additionally, the agents in the studied scenarios are always centered and share a simple appearance, which differ from the settings studied by previous latent action models [3,4]. While it is common to utilize simplified environments for research, the scenarios created by the authors are too unique to verify their applicability to true decision-making scenarios.
- Typical decision-making scenarios, such as Open-X-Embodiment and ProcGen, do not require using the proposed method (the authors confirmed in their response to my rebuttal comment). This also greatly compromises the applicablity of the proposed methods to real applications.
- It would be beneficial to incorporate convincing visualizations like LAPO [3] to demonstrate the distribution of latent actions.

---

[1] Learning Universal Policies via Text-Guided Video Generation

[2] Learning to Act from Actionless Videos through Dense Correspondences

[3] Learning to Act without Actions

[4] Latent Action Pretraining from Videos

**Claims And Evidence:**

Yes, there are supported by clear and convincing evidence.

**Essential References Not Discussed:**

As far as I know, all closely related works are cited appropriately.

**Experimental Designs Or Analyses:**

The paper provides extensive experiments with sufficient details.

**Methods And Evaluation Criteria:**

I have some concerns about how the paper assesses the quality of latent actions. The objective of extracting latent actions is to fully encode action information while minimizing background noise, and the paper utilizes linear probing error to judge the quality of latent actions. However, a lower error does not necessarily equate to higher quality latent actions. Instead, it only indicates that the encoded latent space contains more action information, regardless of the ratio of useful action information.

**Other Comments Or Suggestions:**

Please see the questions below.

**Other Strengths And Weaknesses:**

W1) **The reliability of the metric.** As I mentioned earlier, the quality of latent actions is influenced not only by the amount of action information encoded but also by the ratio of useful information to noise. Therefore, I believe that linear probing error is not a reliable metric for assessing the quality of latent actions, as it only evaluates how much action information are contained in the latent.

W2) **The effectiveness of the method.** Guided by the inappropriate metric, the proposed LAOM removes the information bottlebeck and significantly increase the latent action dimension to 8192. However, according to the downstream performance (blue lines) in Fig.6, these modifications yield little improvement and can even lead to worse performance with 2 and 4 labeled trajectories.

W3) **The scope of the proposed setting.** While introducing action labels mitigates the negative effects of distractors, it may be somewhat unfair compared to the motivation of LAPO (Schmidt & Jiang, 2023). LAPO and its related works exclude action labels during the pretraining stage to ensure their algorithms only rely on videos. This assumption has huge potential to exploit Internet-scale and cross-embodiment data, even without a consistent action format. However, the proposed setting is not aligned with this goal.

**Questions For Authors:**

Q1) **Procgen results without distractors.** Aside from the environments with distractors, is it possible to compare the proposed techniques with LAPO on the original Procgen benchmark to demonstrate their effectiveness? I believe this would help readers fully understand the properties of the proposed method.

**Relation To Broader Scientific Literature:**

The paper aims to improve the effectiveness of action pretraining, which could benefit the development of embodied AI.

**Theoretical Claims:**

I have checked the theoretical claims in this paper.

---

> ### Author Rebuttal · Authors · 2025-04-01
>
> We thank the reviewer for their time and feedback. We have tried to address the concerns below.
>
> > I have some concerns about how the paper assesses the quality of latent actions. The objective of extracting latent actions is to fully encode action information while minimizing background noise, and the paper utilizes linear probing error to judge the quality of latent actions. However, a lower error does not necessarily equate to higher quality latent actions. Instead, it only indicates that the encoded latent space contains more action information, regardless of the ratio of useful action information.
> >
>
> This is indeed a valid concern, which we already discuss in detail in the paper (see Section 4, second column, from lines 269-270). As the reviewer correctly points out, linear probing does not tell us that the resulting latent actions are minimal (in the sense that they contain only information relevant to actions without noise), but only allows us to detect the amount of information about real actions in the latent ones.
>
> It should be noted that the fact that latent actions contain information about real actions is a **necessary condition** for their usefulness. If the latent actions do not contain any information (as expressed by poor probe loss), this automatically means that they are useless for further BC pretraining.  Therefore, before worrying about minimality, it is important to make sure that the latent actions contain real actions at all, which is exactly what we do with our LAOM modifications.
>
> With our experiments in Section 4, we show that in the presence of distractors, naive LAPO (especially with quantization) does not produce latent actions that contain sufficient information about real actions, and thus cannot be used for efficient pre-training. LAOM improves this by a factor of eight, ensuring that latent actions contain real actions, which directly translates into a twofold improvement in return. Does LAOM guarantee latent action minimality? It does not, and we state this explicitly in the paper (first column from line 320). On the contrary, without supervision, LAOM still performs poorly (Figure 6), although we can now hope that supervision will allow us to extract action information from latents, which would be impossible with LAPO (which is clearly illustrated by our main experiment in Figure 1). Demonstrating this was one of our main goals.
>
> > W1) **The reliability of the metric.** … I believe that linear probing error is not a reliable metric for assessing the quality of latent actions, as it only evaluates how much action information are contained in the latent.
> >
>
> We re-emphasise that linear probes were used only to show that LAPO does not work in the presence of distractors, and that even LAOM does not guarantee minimality. Our main contribution and claim is about the need for supervision, and we explicitly demonstrate this in the second part of the paper with experiments in the real environment and with real return. We believe that such clear improvements in return (see Figure 1) clearly indicate the higher quality of latent actions of LAOM vs. LAPO, and LAOM+supervision vs. LAOM.
>
> > W2) **The effectiveness of the method.**
>
> As we stated above (and in responses to other reviewers), linear probing is not our final (and only) metric. It was only used to show that LAPO does not learn good latent actions, and that even after the modifications considered in LAOM - performance did not increase. So the fact that LAOM performs poorly on Figure 6 is not a problem with our approach. On the contrary, this is exactly what we wanted to show in order to highlight the general issues with latent action learning in the presence of distractors. And this is what is greatly improved by the addition of supervision (see Figure 1, 7).
>
> > W3) **The scope of the proposed setting.**
>
> We have to respectfully disagree with the reviewer's assessment. We're not changing the setting in any way, as LAPO, LAPA and other methods still require real actions, we're just suggesting that they be used differently. On the contrary, it is precisely LAPO and related work that considers a simplified setting that lacks the distractors common to real videos, which in turn is not aligned with the goal of pre-training on Internet-scale data. The main purpose of our work was to show that vanilla LAPO will not scale to Internet videos due to distractors.
>
> > Q1) **Procgen results without distractors.**
>
> Our aim was not to provide a state of the art method, but to study the properties of latent action learning methods in the presence of distractors, which is absent in the existing literature. LAOM is more of a set of suggestions for practitioners than a stand-alone method. Therefore, we think that the experiments on ProcGen are out of scope, as they do not allow us to study any additional properties of LAPO in the presence of distractors.

---

> > ### Comment · Reviewer_o8og · 2025-04-03
> >
> > Thank the authors for the detailed rebuttal. I have read all of it, as well as with the comments from other reviewers. Before I update the rating, I still have two major concerns.
> >
> > C1) **Scalability to cross-embodiment web-scale data.** This concern corresponds to W3 in my initial comments. LAPA and its related works can incorporate data from different embodiments to learn a universal policy. However, since the proposed method requires action supervision during training, it raises problems of how to provide supervision when the action formats vary between embodiments and how to ensure the action supervision for one embodiment will not exclude action information for other embodiments (for example, if the objective is to decode the action of the Franka arm, the motion of the WidowX or human could be regarded as distractors, which hurts unified training). The paper studies cross-embodiment training in Section 5, but the four embodiments seem to be always centered and share similar appearances. Compared to previous LAMs, are there any potential limitations if the proposed method is applied to more diverse environments, such as Procgen in LAPO and Open-X-Embodiment in LAPA? If no limitation, could you briefly explain why or provide evidence if applicable?
> >
> > C2) **The actual information predicted by the policy.** As revealed by Figure 10 and the responses to reviewers HSFB and sdKR, the latent capacity is very redundant so that it can pass almost all pixel information through the latent. With the reconstruction loss, the use of action supervision does not actually eliminate distractors. It is possible that the action supervision only makes the output features more suitable for linear probing. If a stronger probing network (instead of a linear layer) is used, the observation embedding could achieve the same MSE loss as the latent actions. As the information for the policy predict is nearly equivalent to original observations, how is the proposed method fundamentally better than predicting observation embeddings, such as UniPi [1] or AVDC [2]? Could you elaborate on this?
> >
> > [1] Learning Universal Policies via Text-Guided Video Generation
> >
> > [2] Learning to Act from Actionless Videos through Dense Correspondences

---

> > > ### Author Response · Authors · 2025-04-08
> > >
> > > > C1) **Scalability to cross-embodiment web-scale data.**
> > >
> > > This is a good question, but we don't think it's quite fair to discuss it only in the context of our work. The constraints you refer to are not unique to our method. The proposition "LAOM requires supervision, but LAPO/LAPA does not, and so LAOM is more limited" may be objected, because formally (which can be easily verified by comparing the losses) LAOM, DynaMo, Moto, IGOR, LAPO, LAPA, Genie, GR00T N1 are all the same method, differing only in the details of the underlying architecture, and so they all have the same limitations. Thus, it is not that "the proposed method requires action supervision during training", but rather that "all latent action learning methods using the LAPO objective require supervision **in the presence of distractors**".
> > >
> > > Our aim was not to propose a state-of-the-art LAM, but to highlight these limitations. As we discussed earlier, LAOM is more of a guide to improving LAPO results in the presence of distractors. However, as we discussed in the paper, without supervision LAM cannot separate noise from control-related features on its own. We show that the reuse of real action labels helps significantly in such a setting.
> > >
> > > We agree that this is a rather restrictive requirement, but without supervision LAM (including LAPO) methods **will not work at all** in the presence of distractors (we demonstrate this in Figure 1). How can we provide supervision if we have no real action labels and/or have multiple unknown action spaces? This is an open question and a very fruitful direction for future research. For example, for egocentric videos, we could use hand tracking as a proxy action to supervise latent action learning. However, we believe that this is currently beyond the scope of our work.
> > >
> > > As for Open-X-Embodiment or ProcGen - there is no need to use supervision as these datasets contain almost no distractors. The need arises when we start using e.g. Ego4D or other real world data such as YouTube.
> > >
> > > We discuss other potential limitations in Appendix A in the current version of the paper.
> > >
> > > > C2) **The actual information predicted by the policy.**
> > >
> > > This is a really important and valid concern, which however criticizes the whole direction of latent action pre-training. Is it true that LAM works better than other methods of pre-training at scale? The honest answer is that we don't know, there are no detailed investigations currently. However, it is gaining popularity as part of foundational models such as the GR00T N1 & AgiBot, so we thought it was crucial to highlight the fundamental limitations of this approach. It is quite possible that in the presence of distractors LAM works no better than any other pre-training method. However, it is important to note that without the improvements suggested in our paper it does not work at all (possibly much worse than UniPi & AVDC).
> > >
> > > As for linear probing, in our final experiments we use MLP with multiple layers as decoder to predict ground-truth actions from BC outputs. With that in mind, we feel that the difference in the final performance reflects how good the latent actions really are. You are right in that even after supervision, there is a lot of redundant information in latent actions, which means there is still a lot of room for improvement in the future.
> > >
> > > We hope we have addressed most of the reviewer's concerns.

---

### Official Review · Reviewer_sdKR · 2025-03-14

**Overall Recommendation:** 4

**Summary:**

This paper focuses on the setting of learning latent actions in the presence of background distractions. The authors investigate improving upon prior latent action pretraining work with recent advances in dynamics and latent action modeling. It shows that multi-step inverse dynamics, large latent action dimension without quantization, forward dynamics in latent space, and augmentations help improve latent action quality when distractors are present, and that supervised training with a small amount of ground truth actions can help with latent action quality and close the gap to behavioral cloning performance under the background distraction setting.

**Claims And Evidence:**

The paper claims that background distractions requires several changes to current latent action inference methods to improve latent action quality and supports it with experimental results.

**Essential References Not Discussed:**

N/A

**Experimental Designs Or Analyses:**

Overall the experiments make sense. However, in this paper, there seems to be no bottlenecks (neither VQ nor dimensional bottleneck) in the latent action. One concern is whether the latent action, as evaluated by downstream policy performance and action probe MSE, is not only a high-dimensional projection of the observation embedding. It would be convincing to see an additional baseline where downstream policy training / action probe MSE is evaluated directly on the observation embedding.

**Methods And Evaluation Criteria:**

The evaluations overall makes sense. The authors test on extracting latent actions with background distractions, and evaluates the latent action quality with linear probe and downstream policy evaluation.

**Other Comments Or Suggestions:**

N/A

**Other Strengths And Weaknesses:**

Strengths: the paper investigates extracting latent actions from video data with background distractors, an important setting to consider for generalizing to Internet-scale videos. It shows that with a few modifications, prior work in latent action pretraining such as LAPO can be improved both in the distractor-free and background distraction settings.
Weaknesses: see experimental designs and questions.

**Questions For Authors:**

- Clarification: is the BC policy also only trained on up to 128 trajectories?
- Clarification: what is the IDM trained on? Is it trained on up to 128 trajectories of ground truth actions, then used to relabel the full dataset for downstream behavioral cloning?
- Baseline: how large is the observation embedding dimension? In Figure 8(b), what is the action probe MSE if you probe directly from the observation embedding?
- Question: why is it the case that LAOM + supervision achieves a low action probe MSE regardless of latent action dimension, and LAOM without supervision action probe MSE depends on the latent action dimension?

**Relation To Broader Scientific Literature:**

The paper develops on recent advances in latent action pretraining such as LAPO, DynaMo, LAPA, etc., which investigates pretraining latent action policies and visual representations from video data. There is also a rich line of work investigating policy robustness under visual distractions. Extracting latent actions from videos is a longstanding important topic for learning robotic policies, as well as dealing with background distractors.

**Theoretical Claims:**

N/A

---

> ### Author Rebuttal · Authors · 2025-04-01
>
> We are grateful to the reviewer for their time, constructive feedback, and suggestions for additional experiments, which we found very valuable. We have tried to answer the questions below.
>
> > Clarification: is the BC policy also only trained on up to 128 trajectories?
> >
>
> Yes, the BC baseline we show in Figures 1, 7, 9 uses the same architecture as the BC in LAM methods, but is simply trained from scratch only on trajectories with available ground truth action labels, from 2 to 128.
>
> We also use a separate BC for normalization on all figures. We pre-train it on full datasets with all action labels revealed to get the maximum possible score with ground truth actions. With such normalization, we can quantify how much performance we have recovered compared to having access to a fully action-labelled dataset.
>
> > Clarification: what is the IDM trained on? Is it trained on up to 128 trajectories of ground truth actions, then used to relabel the full dataset for downstream behavioral cloning?
> >
>
> Yes, this is an accurate description of the overall pipeline.
>
> > Baseline: how large is the observation embedding dimension? In Figure 8(b), what is the action probe MSE if you probe directly from the observation embedding?
> >
>
> Thanks for the suggestion! Due to time constraints, we ran these experiments with three random seeds, but only in the walker environment. Given the previous evidence, we are confident that the results will hold in the remaining environments and will include additional results in the camera-ready version of the paper. We took the observation embedding from the LAOM encoder and trained the linear probe to predict real actions, similar to probing from latent actions. We visualize the results in the following figures [[Figure 1](https://ibb.co/wFRGMhDV), [Figure 2](https://ibb.co/gL6L0zNf)].
>
> As can be seen, for LAOM it is indeed the case that probe from observation embedding is better for smaller latent action dimensionality. This can be explained by the fact that the information bottleneck induces the IDM to mainly encode noise in latent actions, as it can better explain the dynamics (deterministic distractors in the background), while observation embedding mostly preserves the information. At higher latent action dimensions, they are expected to equalize, as latent actions without bottleneck can encode the full dynamics, including noise and real actions. This is exactly the effect we described in Section 4, which motivated us to add supervision.
>
> However, we see a different picture with LAOM+supervision, where the probe from the embedding observation is generally worse than from the latent actions, because with supervision we can ground the latent actions to focus on features relevant for control even with small dimensions, filtering out the noise.
>
> > Question: why is it the case that LAOM + supervision achieves a low action probe MSE regardless of latent action dimension, and LAOM without supervision action probe MSE depends on the latent action dimension?
> >
>
> We believe that in the absence of supervision, as we discuss in Section 4, the information bottleneck is detrimental, as it incentivises the IDM to encode into latent actions a minimum amount of information that is maximally predictive of the next observation. In the case of distractors, this will mostly be noise, as it is easier to predict deterministic videos in the background than actual actions (which also explain much more variation in the overall dynamics). By increasing the latent action dimension, we remove the bottleneck and allow LAOM to encode the full dynamics in actions, including actions but also noise.
>
> On the other hand, LAOM+supervision grounds the latent action space to be predictive of actual actions, which can be much smaller because it does not need to explain noise (actual actions are only ~4-16 dimensions).

---

### Official Review · Reviewer_HSFB · 2025-03-14

**Overall Recommendation:** 4

**Summary:**

- The paper focuses on LAMs, which aim to infer control actions from unlabelled videos
- Here the authors note a benefit of reusing action labels from later in the pipeline to help focus (through supervision) latents on control actions
- This is most effective in the presence of 'distractors', ie non-control action changes
- Empirical results are show on distracting control suite
- A broader investigation of various model design choices and extensions are conducted

**Claims And Evidence:**

Yes, see strengths weaknesses.

**Essential References Not Discussed:**

Missing this, which follows a pipeline similar to LAPA.
- IGOR: Image-GOal Representations are the Atomic Control Units for Foundation Models in Embodied AI

**Experimental Designs Or Analyses:**

See strengths weaknesses.

**Methods And Evaluation Criteria:**

Yes, see strengths weaknesses.

**Other Comments Or Suggestions:**

NA

**Other Strengths And Weaknesses:**

Strengths
- While the problem setting departs from the standard three-stage LAM setup, in my opinion it remains a realistic use-case, where the same set of labels are being used twice in a smart way
- The method is simple
- Overall the method works well, the paper in general directly addresses an issue many LAM users may be interested in
- Good related work description
- Thorough investigation of various aspects of LAMs going beyond the main message provides breadth to the paper (eg fig 8, 9, 10, 12)
- Reasonable eval metrics, normalizing by BC with all labels was a nice touch
- Reasonable baselines, nice to see improvements over the IDM baseline
- Interesting intuition about differences between LAM and IDM approaches (generalization)

Weaknesses
- One major difference with previous work is the decision to open up the bottleneck -- both removing the quantization and moving to a large 8192 dims. This is originally justified by fig 4 and 5, however these only measure MSE -- naturally a wider bottleneck will allow more information about both distractors and actions into the latent, so measuring MSE on actions will reduce. But is having extra distractor information not harmful? Fig 8c partially suggests not.
- In general I'm surprised with this capacity the model doesn't just pipe the entire next observation through the latent -- one possible reason I thought is that the architectures are not powerful enough to unpack all details through the bottleneck, see q about architectures.
- Overall this is not a critical point as it seems to be necessary to get the method to work anyway.
- Some of the architectural choices -- eg latent reconstruction and augmentation -- are becoming pretty standard in other LAM works, so there is limited novelty there, though the paper doesn't overclaim on this aspect anway.
- I'd be interested to dive deeper into the differences between LAM and IDM -- fig 8a suggests the gap closes at some point. But this is probably better left to future work.

Questions
- Could you detail the architecture for IDM FDM and pre-trained latent policy? Also BC baseline. Are these CNNs? In appendix I only saw 'encoder num res blocks'
- Augmentation details -- is there any structure in how you do the random augs? Like do inputs in a sample receive the same crop or?
- Would love to see a zoomed in plot of fig 8 b as can't tell how the MSE changes for the LAOM+supervision.
- For fig 10, what representation is used for the IDM? The final predicted action?
- In fig 8 c, if it seems like the models keep improving with latent dim -- why stop at 8192?

**Questions For Authors:**

See strengths weaknesses.

**Relation To Broader Scientific Literature:**

See strengths weaknesses.

**Theoretical Claims:**

NA

---

> ### Author Rebuttal · Authors · 2025-04-01
>
> We thank the reviewer for their thoughtful and constructive feedback. We address the main questions below.
>
> > Missing this, which follows a pipeline similar to LAPA: IGOR…
> >
>
> We will include the citation, thank you for your suggestion.
>
> > … naturally a wider bottleneck will allow more information about both distractors and actions into the latent …. But is having extra distractor information not harmful?
> >
>
> As we explained in more detail in Section 4, the removal of quantization and the increase in dimensionality is a necessity. Without these changes, latent actions will have no information about real actions at all, and as a consequence will be useless for further pre-training, as demonstrated by the LAPO performance. The best we can hope for in the general case (without supervision) is to encode all dynamics, including noise, but most importantly real actions, into latent actions. That's what the changes in LAOM do. LAOM also does not guarantee that latent actions will be minimal. However, now that we are sure that the real actions are contained in the latent ones, we can hope that with a little supervision we can extract them in a generalizable way. We believe that the results in Figures 1, 7 and 8c clearly show that this is indeed the case.
>
> > I'm surprised with this capacity the model doesn't just pipe the entire next observation through the latent
> >
>
> This is an interesting question that we did not explore in depth as we felt it was beyond the scope of the study. However, we did not observe any evidence of shortcut learning. It is important to note that we used encoders that were not very small for a given task. Given that we are working with 64x64 images, they have enough capacity to predict the next state in pixel space very accurately. As we show in the appendix, the encoders take up about 200M parameters in total.
>
> > I'd be interested to dive deeper into the differences between LAM and IDM
> >
>
> We believe that IDM will perform better than LAM in the limit (e.g. see Figure 9 in [GR00T N1](https://arxiv.org/abs/2503.14734)), but it's also limited to a single action space. However, when the number of labels is very limited, LAM will perform better due to better generalization. Overall, we believe that LAM+supervision combines the best of both worlds with better use of existing action labels.
>
> > Could you detail the architecture for IDM FDM and pre-trained latent policy? Also BC baseline. Are these CNNs? In appendix I only saw 'encoder num res blocks'
> >
>
> We provide some details of the architectures in Appendix D. We use the same visual encoder architecture for all methods (IDM, FMD, BC), which is a simple ResNet borrowed from the open source LAPO code. For FDM we use an identical architecture, swapping conv downsampling with transposed conv upsampling. LAPO, similar to the original code, uses separate encoders in IDM and FDM, while LAOM shares one encoder between them. For LAOM latent FDM we use multiple MLP blocks, inspired by MLP from transformer architecture. To process successive observations, we concatenate images across channels. BC uses ResNet + small action head. The action decoder is a two-layer MLP with a hidden dim of 256.
>
> > Augmentation details
> >
>
> We use several types of augmentations: shift, rotate, change perspective, and combinations such as shift-rotate, rotate-perspective, etc. (as we note in the paper, they are taken from Almuzairee et al., 2024). We sample augmentations for each sample in a batch, but share them across the dimension of the frame stack.
>
> > Would love to see a zoomed in plot of fig 8 b as can't tell how the MSE changes for the LAOM+supervision.
> >
>
> Sorry, here's an enlarged [Figure](https://ibb.co/5X6B5KdH). We will add it to the appendix. Overall, the MSE does not change that much here.
>
> > For fig 10, what representation is used for the IDM?
> >
>
> The IDM can be schematically described as $a = h(f(s_t), f(s_{t+1}))$, where $f$ is a ResNet encoder and $h$ is an action head consisting of MLP. For visualization we used observation embedding after the ResNet encoder, that is $f(s_t)$.
>
> > In fig 8 c, if it seems like the models keep improving with latent dim -- why stop at 8192?
> >
>
> Mostly for practical reasons, since increasing the dimensionality of latent actions increases the overall computational requirements for pre-training. For example, we had to significantly increase the size of the BC in order for it to learn latent actions of dimension 8192 accurately enough (real actions have dimensionality of about 4-16). In fact, to perform with similar quality on real actions, the BC might have been about 4-8 times smaller, because predicting 4 numbers is much easier (LAPA report similar results). To ensure fairness we used larger BC size in all experiments. Our main goal was to show a trend, so we did not see the need to go further.

---

> > ### Comment · Reviewer_HSFB · 2025-04-03
> >
> > Thank you for this response. For now I maintain that it is a strong paper and is worth presenting to ICML attendees -- will continue discussion with reveiwers as needed.

---

### Decision · Program_Chairs · 2025-05-01

**Decision:**

Accept (poster)

**Comment:**

This paper received generally positive reviews for its empirical investigation into latent action learning in the presence of distractors. Reviewers appreciated the realistic problem setting, clear motivation, and thorough experimental analysis. The work demonstrates that small amounts of action supervision and design changes—such as increasing latent dimensionality and incorporating forward dynamics—can significantly improve performance in distractor-heavy environments.

While the method shows solid empirical gains, one reviewer expressed strong reservations regarding the scalability and real-world applicability of the setting. Concerns include the lack of strong comparisons to prior video prediction models, limited generalizability beyond synthetic distractions, and the potential misalignment with the broader goals of actionless learning. Additionally, some questioned the informativeness of the chosen metrics and the lack of evaluation on more diverse benchmarks like Procgen.

Despite these limitations, the paper offers valuable insights into a challenging and under-explored aspect of latent action learning. The AC finds the contributions relevant and timely and recommends acceptance, while encouraging the authors to better contextualize their setting and discuss broader applicability in the final version.